# Visuo-Tactile Transformers for Manipulation

**Yizhou Chen**     **Andrea Sipos**     **Mark Van der Merwe**     **Nima Fazeli**
Department of Robotics, University of Michigan
Ann Arbor, MI 48109, United States
{yizhouch, asipos, markvdm, nfz}@umich.edu
https://www.mmintlab.com/vtt

**Abstract:** Learning representations in the joint domain of vision and touch can improve manipulation dexterity, robustness, and sample-complexity by exploiting mutual information and complementary cues. Here, we present Visuo-Tactile Transformers (VTTs), a novel multimodal representation learning approach suited for model-based reinforcement learning and planning. Our approach extends the Visual Transformer [1] to handle visuo-tactile feedback. Specifically, VTT uses tactile feedback together with self and cross-modal attention to build latent heatmap representations that focus attention on important task features in the visual domain. We demonstrate the efficacy of VTT for representation learning with a comparative evaluation against baselines on four simulated robot tasks and one real world block pushing task. We conduct an ablation study over the components of VTT to highlight the importance of cross-modality in representation learning.

**Keywords:** Multimodal Learning, Reinforcement Learning, Manipulation

## 1 Introduction

Deep reinforcement learning (RL) algorithms have solved numerous challenging tasks from raw observations (e.g., images) [2, 3, 4, 5]. Despite these successes, learning directly from such high-dimensional observation spaces can be numerically difficult/unstable, sensitive to hyper-parameters, and sample inefficient [4]. These challenges have limited the utility of these methods for robotics applications because data collection is time intensive, costly, and task variance is high.

To address this, recent research in representation learning aims to build compact "latent" representations of the underlying states [4, 6, 7, 8, 9, 10]. These compact representations are typically trained together with a policy using prediction and reconstruction losses along with task-dependant rewards. The policy then learns from the low-dimensional latent representation rather than the high-dimensional raw sensory observation. This approach offers stability and sample-efficiency for policy learning owing to the information-dense latent representations generated from rich prediction and reconstruction supervisory signals. As such, latent representations are a promising tool for real-world robotic planning and RL.

Most research in robot latent representation learning has focused on images and proprioception [2, 5, 11, 12, 13, 14, 15]. The sense of touch, together with vision, plays an important role in robotic manipulation; however, visuo-tactile representation learning is still in its relative nascency. Recent works [10, 16, 17, 18, 19, 20, 21, 22] have proposed visuo-tactile methods; however, the latent representations used are typically vectors or clusters in $\mathbb{R}^n$ and do not exploit attention architectures [23]. These representations can smooth over fine details and struggle with locality in images. These issues are particularly troublesome for manipulation as the difference between in-contact and out-of-contact can be small in pixel space or the robot may need to move significantly to contact objects.

In this paper, we propose a novel latent representation learning method that addresses these challenges. Our approach extends the popular Visual Transformer (ViT) [1] to integrate multimodal feedback from vision and touch – the Visuo-Tactile Transformer (VTT). Our approach focuses spatially-aware attention on important visual task features using tactile feedback. We show how this cross-modal attention creates a rich spatially distributed latent space that facilitates more efficient task execution. We evaluate the efficacy of our approach on four simulated and one real-world contact-rich interactions, benchmarking against state-of-the-art techniques.

6th Conference on Robot Learning (CoRL 2022), Auckland, New Zealand.

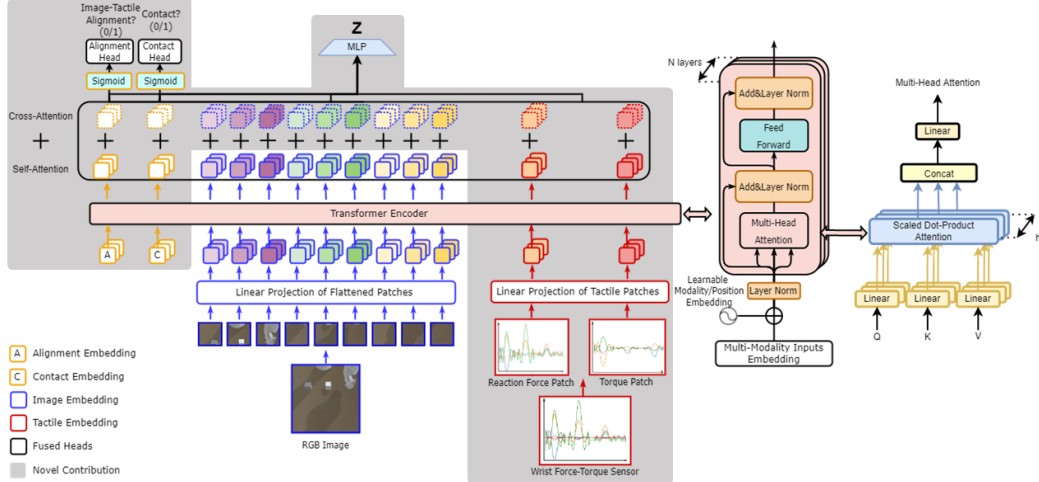

Figure 1: **Architecture:** Schematic of our proposed Visuo-Tactile Transformer. Parts of the diagram with gray backgrounds are novel contributions and differentiate our structure from ViT.

## 2 Related Work

The learning community has made significant progress in latent space dynamics learning, demonstrating promise in relatively low-dimensional [24, 25, 26, 27] and high-dimensional [4, 6, 7, 8, 9] spaces. The key idea in these approaches is to learn a latent dynamics model that compactly represents the high-dimensional observation space that is then used for planning or model-based RL. Many of these approaches consider continuous control tasks that may be applicable to the robotics domain [28, 29, 30, 31, 32, 33]. Existing approaches focus on learning latent dynamics from visual feedback. Here, our work explicitly addresses multimodal learning in the visuo-tactile domain.

The robot learning community has also made progress in learning latent space dynamics. In particular, cross-modal learning has recently gained traction [10, 16, 17, 18, 19, 20, 21, 22] where the current trend is in visuo-tactile representations. Our work is most similar to [10, 16] and we use their encoders as our baselines. The key difference between these works and ours is the latent representation. Rather than using a vector in $\mathbb{R}^n$, VTT learns a multimodal latent heatmap where attention is distributed spatially. We hypothesize that VTT's explicitly spatially-aware cross-attention is more effective for representation learning in terms of sample efficiency and expected reward than either of these approaches. We evaluate this claim by benchmarking against existing approaches.

Our proposed approach is an extension of the Transformer [23] and ViT [1] to the visio-tactile domain. Recent research has demonstrated the efficacy of attention mechanisms for sequence-to-sequence and spatially-distributed inputs. A tighter integration of encoders developed by the computer vision community [34, 35, 36] with robotic applications can lead to significant advances in representation learning that enable efficient planning and RL.

## 3 Visuo-Tactile Transformers

At a high level, VTT produces compact latent representations by fusing visuo-tactile inputs with self and cross-modal attention mechanisms. This enables robotic agents to leverage the complementary nature of these sensing modalities for policy learning. Additionally, VTT supplements image and tactile input patches with learned contact, alignment, and position/modality embeddings to further improve multi-modal reasoning. The overall VTT architecture is shown in Fig. 1. We present the details of the architecture and its components in the following sections.

## 3.1 Modality Patches

Before sending inputs to the transformer encoder to compute self and cross-modal attention, both the image and tactile inputs need to be patched. RGB image inputs are patched and embedded with one 2-D convolution layer as in ViT [1]. Tactile inputs from a wrist-mounted force-torque sensor are patched into reaction forces and torques, then a linear projection is applied to embed them. Image and tactile modalities are notated $I$ and $T$ respectively. We denote the embedded modality patch as $X_M \in \mathbb{R}^{(P_I+P_T) \times d}$, where $P$ is the number of patches in each modality and $d$ is the number of features in each modality, which we assume to be identical. This embedded modality patch $X_M$ can be written $[X_I; X_T]$ and is used as input to the first attention layer, as shown in Fig. 1.

## 3.2 Self and Cross-Modal Attention

We implement self and cross-modal attention mechanisms by adapting the multi-head attention method from Transformer architecture [23]. This adaptation is shown on the right side of Fig. 1. The transformer encoder is composed of a stack of $N$ identical attention layers. Each attention layer has two sub-layers: a multi-head attention sub-layer and a feed-forward sub-layer. When $1 \leq n+1 \leq N$, the output $X_n$ of the $n^{\text{th}}$ attention layer $A_n$ will be the input for the $(n+1)^{\text{th}}$ attention layer in the transformer encoder. To show how self and cross-modal attention are computed through $N$ attention fusion layers, we first analyze $A_{n=1}$ and then generalize to $n > 1$ layers.

To calculate the self and cross-modal attention at layer $n = 1$, we first project the post layer normalization ($LN$) of embedded modality patch $X_M$ into the Query ($Q_{n=1}^i$), Key ($K_{n=1}^i$), and Value ($V_{n=1}^i$) shown in Eq. 1, where $h$ is the number of attention heads and $1 \leq i \leq h$.

$$Q_{n=1}^i = LN \begin{bmatrix} X_I \\ X_T \end{bmatrix} W_Q^i, \quad K_{n=1}^i = LN \begin{bmatrix} X_I \\ X_T \end{bmatrix} W_K^i, \quad V_{n=1}^i = LN \begin{bmatrix} X_I \\ X_T \end{bmatrix} W_V^i \tag{1}$$

In this formulation, $W_Q^i \in \mathbb{R}^{d \times d_K}$, $W_K^i \in \mathbb{R}^{d \times d_K}$, and $W_V^i \in \mathbb{R}^{d_K \times \frac{d}{h}}$ are weights for Query, Key and Value, where $d$ is defined in Sec. 3.1 and $d_K$ is the dimension of features in the Key. Following Eq. 1, we note that the Query, Key, and Value can be written as

$$Q_{n=1}^i = \begin{bmatrix} Q_I \\ Q_T \end{bmatrix}_{n=1}^i, \quad K_{n=1}^i = \begin{bmatrix} K_I \\ K_T \end{bmatrix}_{n=1}^i, \quad V_{n=1}^i = \begin{bmatrix} V_I \\ V_T \end{bmatrix}_{n=1}^i \tag{2}$$

After projecting the embedded modality patch $X_M$ into $Q_{n=1}$, $K_{n=1}$, and $V_{n=1}$, we use Eq. 3 to calculate the attention at each head $i$ using a scaled dot product with regularization factor $a = \sqrt{d}$.

$$A_{n=1}^i = Atten(X_{n-1} = X_M)^i = softmax\left(\frac{Q_{n=1}^i (K_{n=1}^i)^T}{a}\right) V_{n=1}^i \tag{3}$$

For clarity and brevity, we expand Eq. 3 in terms of $Q$, $K$, and $V$ to isolate self and cross-modal attention.

$$A_{n=1}^i \sim QK^T V = \underbrace{\begin{bmatrix} Q_I K_I & Q_I K_T \\ Q_T K_I & Q_T K_T \end{bmatrix}}_{\text{Attention Heatmap}} \begin{bmatrix} V_I \\ V_T \end{bmatrix} = \underbrace{\begin{bmatrix} Q_I K_I V_I \\ Q_T K_T V_T \end{bmatrix}^i}_{\text{Self Attention}} + \underbrace{\begin{bmatrix} Q_I K_T V_T \\ Q_T K_I V_I \end{bmatrix}^i}_{\text{Cross Attention}} \tag{4}$$

From Eq. 4 we can see that the attention heatmap is constructed with image and tactile self attentions $Q_I K_I$ and $Q_T K_T$ on the diagonal and cross-modal attentions $Q_I K_T$ and $Q_T K_I$ in the upper and lower triangles. The attention output of Eq. 4 is the sum of self and cross-modal attention. Finally, the multi-head attention $A_{n=1}$ is formed by concatenating each attention head as shown in Eq. 5. Multi-head attention encourages the model to explore the attention subspace.

$$A_{n=1} = [A_{n=1}^1, A_{n=1}^2, ..., A_{n=1}^h] \in \mathbb{R}^{(P_I+P_T) \times d} \tag{5}$$

To connect the sub-layers in each attention layer, we implement the residual connection method from [37]. Finally, the output $X_{n=1}$ of the attention layer $A_{n=1}$ can be written

$$X_{n=1} = f_{n=1}(A_{n=1}, X_{n-1} = X_M) \tag{6}$$

where $f_{n=1}$ is a learned nonlinear function induced by nonlinear activation functions and operations.

To generalize to $n > 1$ attention fusion layers, $X_{n-1}$ is the input for attention layer $A_n$ and is projected into $Q_n$, $K_n$ and $V_n$ using Eq. 1. This proposed attention mixture allows the agent to iteratively leverage weights of self and cross-modal attention.

### 3.3 Learned Embeddings

To improve feature learning for RL policies, we propose three learnable embeddings to bolster the modality patches and attention mechanisms we introduced in Sec. 3.1 and Sec. 3.2. We evaluate effects of these embeddings on model performance in the ablation study presented in Sec. 4.2.

**Contact Embedding:** Intuitively, contact embeddings are used to pull latent codes for alike contact states together while pushing latent codes for dissimilar contact states apart. After $N$ layers, the contact embedding $X_C \in \mathbb{R}^{1 \times d}$ becomes the contact head $C_{head} \in \mathbb{R}^{1 \times d}$ following the same method outlined in Sec. 3.2. We form a contact loss in Eq. 7 by using $C_{head}$ to predict the contact state of the system then comparing that prediction to the ground truth $C_{gt}$.

**Alignment Embedding:** Although contact embeddings introduce binary contact recognition, temporal alignment of tactile signatures to images is non-trivial [10, 16]. Alignment embeddings shape the latent space to seek similarities in temporally aligned modalities and differences in temporally misaligned modalities [10, 38]. Therefore, alignment embedding $X_{Al} \in \mathbb{R}^{1 \times d}$ is introduced and becomes the alignment head $Al_{head} \in \mathbb{R}^{1 \times d}$ after $N$ layers. As in the previous section, we form a loss in Eq. 7 by using $Al_{head}$ to predict whether the tactile and image data are aligned then comparing that prediction to the ground truth $Al_{gt}$.

The contact and alignment losses are defined by a binary cross entropy with logits loss $BCE_{logits}(\cdot)$.

$$\ell_{VTT} = BCE_{logits}(MLP(Al_{head})), Al_{gt}) + BCE_{logits}(MLP(C_{head}), C_{gt}) \tag{7}$$

**Position/Modality Embedding:** The final type of learnable embedding we implement is a position/modality embedding. We found that $[X_C; X_{Al}; X_M]$ do not contain the position or modality of information, so it is necessary to include position/modality embedding $X_P \in \mathbb{R}^{1 \times (2+P_I+P_T)}$. Inspired by [1, 39], we add this embedding to our structure to form $[X_C; X_{Al}; X_M] + X_P$.

### 3.4 Discussion of Compressed Representation Head

The output $X_N$ of the transformer encoder is a series of fused heads originating from the visuo-tactile embedded patches $X_M$, contact embeddings $X_C$, alignment embeddings $X_{Al}$, and position/modality embeddings $X_P$. In order to use our learned representation from VTT for RL, we need to think about the dimensionality of the latent vector $\mathbf{z}$. If we use $X_N \in \mathbb{R}^{(2+P_I+P_T) \times d}$ directly, $\mathbf{z}$ will be too high-dimensional. Consequently, we propose compressing all fused heads from $\mathbb{R}^{(2+P_I+P_T) \times d}$ to $\mathbb{R}^{(2+P_I+P_T) \times \frac{d}{c}}$ with constant $c > 4$ by using an MLP. This compression results in $\mathbf{z} \in \mathbb{R}^{1 \times \frac{d}{c}(2+P_I+P_T)}$. Our proposed method will allow image and tactile attention heads from VTT to be used for latent dynamics learning while introducing minimum inductive bias.

### 3.5 Combining with Reinforcement Learning

Due to the frequency of visual occlusions and high levels of tactile perception uncertainty in our manipulation benchmarks, we formulate our manipulation tasks as partially observable Markov decision processes (POMDPs) [40]. There have been a number of advances in model-based RL for POMDPs [41, 4, 42, 2]. From these, we chose the Stochastic Latent Actor Critic (SLAC) [4] algorithm because of its stability and robustness. These characteristics accommodate high stochasticity and provide an effective baseline method to compare visuo-tactile fusion mechanisms.

SLAC is composed of a model-learning component and a policy-learning component. The model-learning piece is built with a factorized sequential variational autoencoder (VAE) $f_\theta$ [43]. $f_\theta$ uses a Gaussian distribution to form a prior model $p$ and a posterior model $q$. The prior model $p(\mathbf{z}_t^d | \mathbf{z}_{t-1}^d, a_{t-1})$ propagates latent dynamics from an action $a$ and the posterior model $q(\mathbf{z}_t^d | \mathbf{z}_{t-1}^d, \mathbf{z}_{t-1}, a_{t-1})$ integrates observations. In this notation, $\mathbf{z}_t^d \sim f_\theta(\mathbf{z}_{0:t}, \mathbf{z}_{0:t-1}^d, a_{0:t-1})$. To close the distribution gap between the prior and posterior models, KL divergence is introduced. Finally, $\mathbf{z}_t^d$ is decoded to reconstruct the visual input $O$ and infer a reward $r$. VTT losses from Eq. 7 are composed with SLAC losses to form the total model-learning losses in Eq. 8.

$$\ell_{model} = \ell(O_t | \mathbf{z}_t^d, a_{t-1}) + \ell(r_t | \mathbf{z}_t^d, a_{t-1}) + \ell_{KL}(q||p) + \ell_{VTT} \tag{8}$$

The policy-learning component is implemented using the soft actor critic (SAC) algorithm [44]. SAC is an off-policy actor-critic deep RL algorithm that maximizes future return and entropy for exploration. For further details and derivation on the model and policy learning, see [44, 4]. It is important to note that the latent representation $\mathbf{z}$ from VTT is utilized by both components. Additionally, the critic's value function loss is backpropagated through VTT to the attention mechanism.

# 4 Experiments and Results

In this section, we benchmark VTT's performance against two popular multi-modality fusion techniques on four simulated robot manipulation tasks. Next, we present an ablation study that evaluates the importance of our contact loss and alignment loss on simulation tasks. We then evaluate VTT on a real-world pushing task. Finally, we visualize the attention maps produced by VTT on simulation and real-world tasks to illustrate the effects and intuition of cross-modal attention in the latent space.

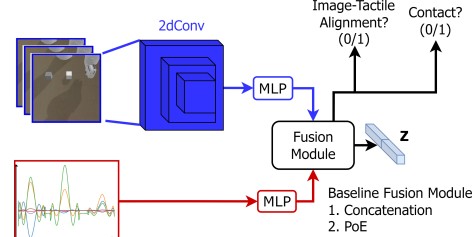

Figure 2: **Baseline Structure**: We compare VTT to concatenation and product of experts (PoE) baselines. These methods are inserted at the 'fusion module'.

## 4.1 Simulated Experiments

**Baseline Structures:** To evaluate the performance of VTT, we benchmark against two popular multi-modality fusion methods: concatenation and product of experts (PoE) [10, 16]. Fig. 2 shows the architecture of these baselines, where the fusion model is the key distinction between the methods. Both baselines follow the same training pipeline as VTT to allow for fair comparisons. We denote the inputs of the fusion module as $E_I \in \mathbb{R}^{1 \times d_I}$ and $E_T \in \mathbb{R}^{1 \times d_T}$.

In the concatenation fusion module, the latent vector $\mathbf{z}$ becomes $\mathbf{z} \in \mathbb{R}^{1 \times (d_I + d_T)} = [E_I; E_T]$. However, the PoE fusion module treats $E_I$ and $E_T$ independently and applies a VAE-like structure to each latent code individually. $E_I$ and $E_T$ pass through separate MLPs to generate separate means $(\mu_{E_I}, \mu_{E_T})$ and variances $(\sigma_{E_I}, \sigma_{E_T})$ that are mapped to a normal distribution using KL divergence, then fused using PoE as shown in Eq. 9 for $i$ modalities and $j$ features. The latent space in this method is also shaped by contact and alignment classification. Further details about each of these baselines can be found in their published works.

$$\sigma_j^2 = \left( \sum_{i=1}^{2} \sigma_{ij}^2 \right)^{-1}, \mu_j = \left( \sum_{i=1}^{2} \frac{\mu_{ij}}{\sigma_{ij}^2} \right) \left( \sum_{i=1}^{2} \sigma_{ij}^2 \right)^{-1} \tag{9}$$

**Simulation Experiment Setup:** We conduct simulated experiments of four manipulation tasks in Pybullet. The selected tasks are Pushing, Door-Open, Picking, and Peg-Insertion. Based on the continuous control manipulation benchmark developed by ElementAI [45], we construct a dense reward setting for these tasks. The agent receives an $84 \times 84 \times 3$ RGB image and a $1 \times 6$ wrist reaction wrench. Task rewards and parameters are outlined below.

**1. Pushing** Push a white block to a grey target pose. The agent receives +25 reward for finishing the task. In this task, we vary the initial pose and mass of the block as well as the target pose.

**2. Door-Open** Open a door from $\theta_{door} = 0$ to $\theta_{goal}$. When $\theta_{door} < \theta_{goal}$, we penalize the agent $-\lambda(\theta_{goal} - \theta_{door})$. When $\theta_{goal} = \theta_{goal}$, the agent receives +25 reward. We incorporate curriculum learning [46] into this task by randomly initializing $\theta_{door} > 0$ during training only. In this task, we vary the color of the door and frame.

**3. Picking** Pick up a white block. We force the gripper to touch the block with a location-based $-\lambda(|Loc_{gripper} - Loc_{block}|)$ penalty. The agent receives +1 reward for grasping when the wrist feels significant downward force. The agent receives +25 reward when the block is lifted to a certain height. In this task, we vary the initial pose, mass, and size of the block.

**4. Peg-Insertion** Insert a peg into a hole. We force peg-hole alignment with a location-based $-\lambda(|Loc_{peg} - Loc_{hole}|)$ penalty. To encourage the agent to lower the peg into the hole, the agent receives $+\lambda(|z_{max} - z_{peg}|)$ reward after alignment. The agent receives +25 reward when the peg in fully inserted. In this task, we vary the color of the peg and the color of the hole.

**Training Details:** We follow the SLAC training schedule to pretrain the dynamics model $f_\theta$ and fusion modules VTT, concatenation, and PoE. After the pretraining, SLAC and VTT are trained together. For the model we use $lr = 0.0001$ and batch size 32. For the policy we use $lr = 0.0003$ and batch size 256. We train using the Adam optimizer [47]. Further implementation details are included in the supplementary material.

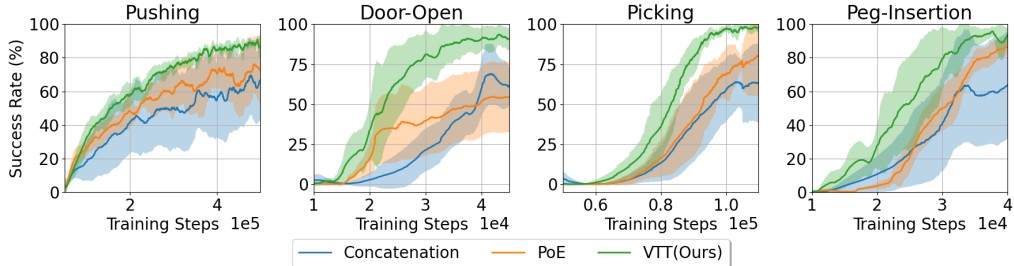

Figure 4: **Simulation Results:** We compare the performance of our method to baselines across 8 trials and find that VTT outperforms the baselines in all but one task. The variance of VTT's learning curve is significantly smaller than either baseline, suggesting better numerical stability of VTT.

**Baseline Comparison:** We compared VTT with the concatenation and PoE baselines over 8 trials. The results for each task with each fusion module are shown in Fig. 4. Compared to the baselines, our proposed approach achieves higher sample efficiency on all tasks and higher success rate in all but the Peg-Insertion task. We speculate that this task may not require as much spatial reasoning as the other tasks because the robot is always grasping the peg and only has to make contact with the table then insert. The remaining tasks all require the robot to first reach for and grasp/push an object, then perform the remainder of the task. Additionally, our approach shows lower variance with respect to the baselines. This suggests that the latent representation is numerically easier and more stable to learn.

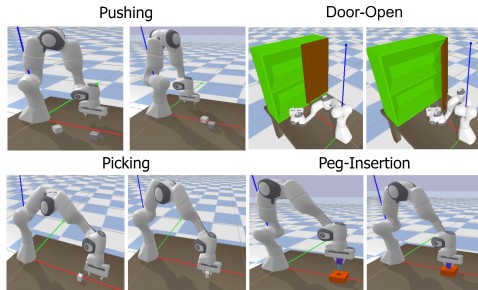

Figure 3: **Manipulation Tasks**: We evaluate VTT on four tasks in Pybullet. We vary visual and physical parameters in each task.

## 4.2 Ablation Studies

In this section, we conduct three ablation studies to verify the fidelity of our method. Firstly, we ablate the contact and alignment losses in the VTT loss function. Secondly, we ablate the compression of latent vector **z**. Finally, we ablate model sizes by adjusting parameter counts (Appendix Sec. 6.4).

### 4.2.1 Loss Function

To show the importance of contact and alignment losses in VTT, we conduct an ablation study by sequentially removing the contact and alignment embeddings and losses and performing the Door-Open and Peg-Insertion tasks. Overall, our results in Fig. 5a indicate that each of these methods is critical for achieving high sampling efficiency and success rate. The lack of contact and alignment losses can cause miscorrelation of images and tactile feedback. The representation induced by such miscorrelations may mislead the agent's state value estimation and decision making, leading to worse overall performance. Additionally, we notice that the agent performs worse without the contact loss than without the alignment loss. This behavior is likely due to the contact-rich tasks we use to evaluate our method. One potentially useful application of this misalignment may be as a metric for evaluating just how out of distribution a novel observation is.

### 4.2.2 Latent Vector Compression

To investigate the effect of compression and the amount of compression, we conducted an ablation study. We find that our chosen compression value of $c = 12$ yielded the best results in the Door-Open and Peg-Insertion tasks. This level of compression allowed VTT to learn faster and yield generally better performance than other compression values, as shown in Fig. 6.

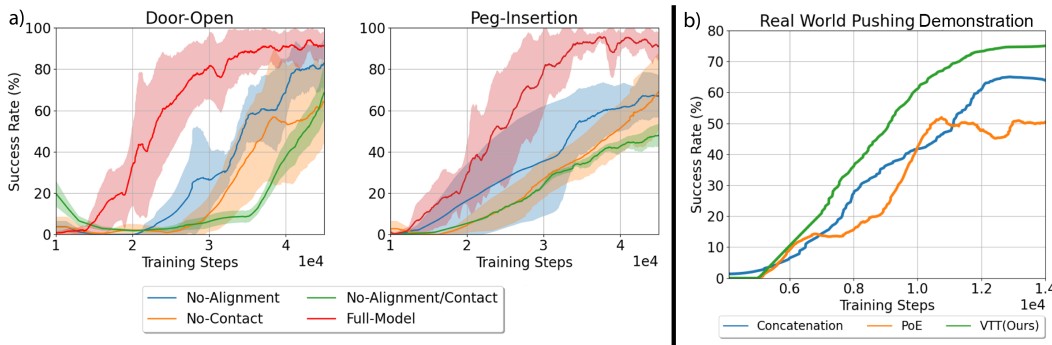

Figure 5: **a) Ablation Results:** We sequentially remove the contact and alignment losses from VTT to show performance in two simulated tasks without them. We find that the full method outperforms each partial method. **b) Real-World Demonstration:** We find that VTT also outperformed the baselines in the real-world Pushing task in terms of sample efficiency and success rate.

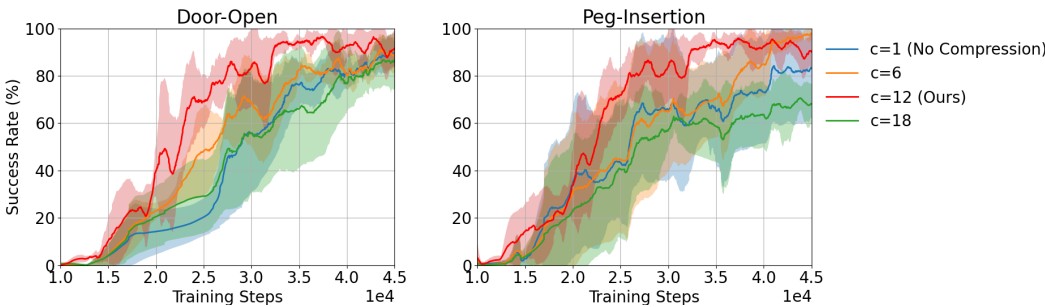

Figure 6: **Latent Vector Compression:** We examine how compression of the latent vector **z** impacts the performance of VTT and find that a value of $c = 12$ yields the best results.

### 4.3 Real-World Demonstration

**Real-World Demonstration Setup:** We conduct the Pushing task in the real world using a Franka Emika Panda robot arm with a wrist-mounted ATI Gamma F/T sensor and custom end-effector. The block is a 2.25-in steel cube with Apriltags on its visible sides. These tags are tracked by two Intel RealSense D435 cameras located in front of the robot (Camera 1) and to the back right of the robot (Camera 2). We use the RGB image from Camera 1 as input $E_I$ and utilize Camera 2 to track the block in case of occlusion. We use the reaction force and torque from the F/T sensor as input $E_T$. This setup is shown in Fig. 7. The agent will receive +25 reward for finishing the task and $-10|Loc_{block} - Loc_{goal}|$ penalty when the task is not finished. Each trial has a 100 step maximum.

**Real-World Baseline Comparison:** Similarly to the simulated tasks, we evaluated VTT against the concatenation and PoE baselines. As in the simulation results, our approach outperforms the other fusion modules in the real world as shown in Fig. 5.

### 4.4 Attention Map Visualization

We illustrate the output of VTT's attention mechanism in Fig. 8 for simulated Pushing and Door-Open tasks. These heatmaps are the average of the multi-head attention from Eq. 4. We empirically observe 2 distinct phases: contact-free and in-contact. In contact-free steps, we observe the attention heatmap highlight the robot and the object (Pushing $t = 1$, Door-Open $t = 2$). This is likely due to correlation between the goal and the change in the tactile signature that arises from contact between end-effector and object. Once contact is present, the attention focuses mostly on the contact interaction and the goal. Again, this is likely due to the correlation between the reward signal and

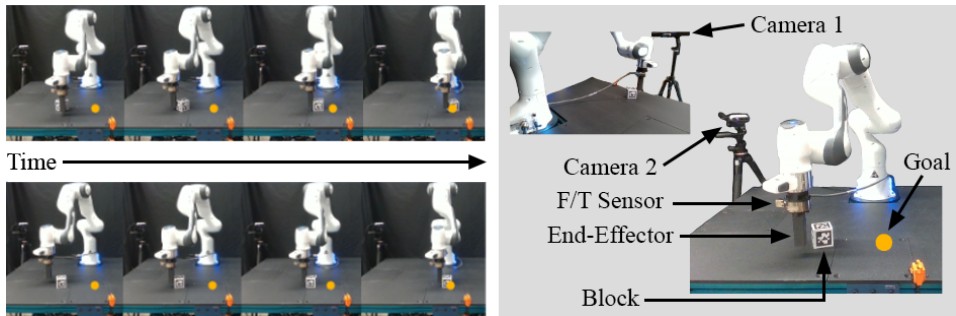

Figure 7: **Real-World Demonstration Setup:** In the left pane we show two successful block pushing tasks. In the right pane we note the location of our cameras and F/T sensor, block, and goal.

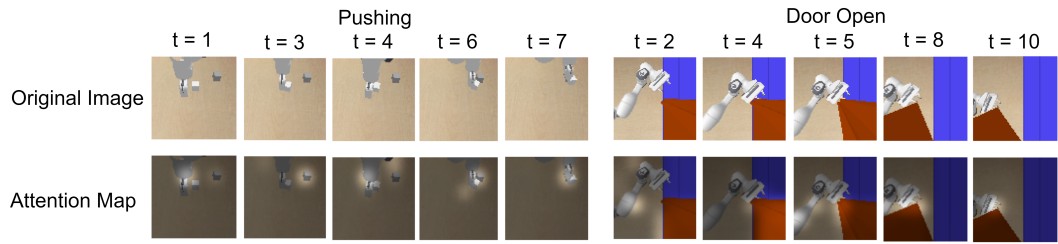

Figure 8: **Attention Heatmap:** We overlay the attention heatmaps from Eq. 4 onto RGB inputs. Further visualizations are available in Sec. 6.3 along with plots of visual and tactile attention at the timesteps visualized. Additional examples are shown in Figs. 9 and 10 (Sim and Real-world).

the tactile signature. We note that the object and the goal can be far from the robot in the visual scene, yet our attention mechanism is able to correctly identify these key features.

## 5    Discussion and Limitations

We expect that VTT will enable important downstream applications such as visuo-tactile curiosity. Curiosity [48] makes exploration more effective by maximizing a proxy reward. This reward is often formulated as a series of actions that increase representation entropy. Similarly to [45], VTT can measure potential mismatch in vision and touch and use this score to explore. VTT advantageously incorporates spatial attention to localize regions of interest in an image, which sets it apart from other multi-modal representation methods. Despite its promise, the current iteration of VTT has a few limitations. Firstly, our implementation uses 6-dimensional reaction wrenches from F/T sensors as tactile feedback. This low-dimensional signal can often alias important contact events and is generally not as informative as collocated tactile sensing like [49, 50]. To address this, the tactile patching and embedding need to be adapted to handle high-dimensional tactile signatures. Secondly, we only evaluated VTT on rigid-body interactions. This choice was motivated by the availability of simulation platforms that incorporate both tactile feedback and deformables and that are fast enough for RL applications. It is not entirely clear how VTT will handle deformables; though we suspect that attention will be distributed over both the deformation and the contact region.

**Acknowledgments**

The authors would like to gratefully thank reviewers for giving useful comments. This material is based upon work supported by the National Science Foundation Graduate Research Fellowship Program under Grant No. 1841052. Any opinions, findings, and conclusions or recommendations expressed in this material are those of the authors and do not necessarily reflect the views of the National Science Foundation.

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

# 6 Appendix

## 6.1 Implementation Details of VTT

With reference to the notation in Sec. 3 and Fig. 1, VTT uses one 2-D convolution layer to patch an $84 \times 84 \times 3$ image into $X_I \in \mathbb{R}^{36 \times 3}$. Each patches goes through one linear projection layer to make $X_I \in \mathbb{R}^{36 \times 384}$. In the real-world experiment, we reduce all the channel sizes to 128. Prior to patching, VTT separates the $1 \times 6$ wrist tactile feedback into $1 \times 3$ reaction force and $1 \times 3$ torque. Features are extracted through one linear projection layer to form $X_T \in \mathbb{R}^{2 \times 384}$. Contact embedding $X_C \in \mathbb{R}^{1 \times 384}$, alignment embedding $X_A \in \mathbb{R}^{1 \times 384}$, and position embedding $X_P \in \mathbb{R}^{40 \times 384}$ are initialized with weights based on a truncated normal distribution by following the class embedding initialization in [1]. We note here that $d_I = d_T = d_A = d_C$ is required for patch-wise concatenation.

$W_Q, W_K, W_V$ are formed with 2-layer MLPs. Each MLP is constructed with two $384 \times 384$ linear layers and has GELUs as activation functions after each linear layer. The feed forward block is also constructed with the same MLP. We chose the attention head number $h = 8$ to effectively spread attention distribution and $N = 6$ attention fusion iterations.

The contact and alignment heads from the transformer encoder's output, denoted $C_{head}$ and $Al_{head}$, are passed through a 1-layer MLP. The MLP is constructed with one $384 \times 1$ linear layer followed by sigmoid activation function for alignment/contact binary classification. 1 is assigned for positive pairing/contact, 0 is assigned for negative pairing/non-contact. When feeding negative pairs, RL's parameters are frozen to prevent misleading gradients for policy learning. As in [10, 16], the alignment embedding is used alongside a training scheme that samples both aligned data and temporally misaligned data.

The fused heads that are output from the transformer encoder $F_{head} \in \mathbb{R}^{40 \times 384}$ have dimensions that are too large for model-based RL. Therefore, each fused head is compressed by a 2-layer MLP constructed with $384 \times 40 \times 32$ linear layers followed with RELU activation functions amid each linear layer. The resulting $A_N \in \mathbb{R}^{40 \times 32}$ are flattened and passed through the 2-layer MLP to form $\mathbf{z} \in \mathbb{R}^{1 \times 288}$.

## 6.2 Further Details of VTT with SLAC

With reference to notation in Sec. 3, the main objective of SAC is to maximize entropy of policy $\mathcal{H}(\pi_\phi(\cdot|a_{t-1}, \mathbf{z}_t))$ while also maximize following POMDP based actor-critic objective functions:

$$J_Q(\beta) = \mathop{\mathbb{E}}_{\mathbf{z}_{1:\tau+1}^d \sim q} \left[ \frac{1}{2} (Q_\beta(\mathbf{z}_\tau^d, a_\tau) - (r_\tau + \gamma V_{\bar{\beta}}(\mathbf{z}_{\tau+1}^d)))^2 \right]$$

$$J_\pi(\zeta) = \mathop{\mathbb{E}}_{\mathbf{z}_{1:\tau+1}^d \sim q} \left[ \mathop{\mathbb{E}}_{a_\tau \sim \pi_\zeta} [\alpha log(\pi_\zeta(a_{\tau+1}|\mathbf{z}_{1:\tau+1}, a_{1:\tau})) - Q_\beta(\mathbf{z}_{\tau+1}^d, a_{\tau+1})]] \tag{10}$$

Where $\beta$ is the parameter of soft Q-function, and $\bar{\beta}$ is delay of $\beta$ for state value V-function. $\zeta$ is parameter of policy. Unlike typical model-based RL, instead of using belief representation, SLAC's policy learning is only conditioned on the past and current $\tau + 1$ steps. The hyperparameters of SLAC follow its published code.

## 6.3 Further Attention Visualization

In addition to visualizing the attention on the RGB inputs, we plot the proportion of attention between visual and tactile inputs over time during the simulated Door-Open task and the real-world Pushing task. We find that at the beginning of the task the attention is dominated by vision in both simulation and the real world. Upon instances of contact, we find that the attention heatmap contracts to the pixel-space of the RGB input where contact is located. At the same time, the proportion of tactile attention increases. In some cases, such as in Row 4 of Fig. 10 (green/green), the modalities meet in the middle near a 50/50 split of attention. We find that in the real-world task, the attention is slightly more balanced at the start of the task and again becomes evenly split upon contact. In some real-world examples, the tactile attention spikes above the visual attention immediately after contact then becomes balanced around 50/50 (Row 1, Row 2). We find this result to be unsurprising because it is in line with our intuition about the nature of multimodality in contact-rich settings such

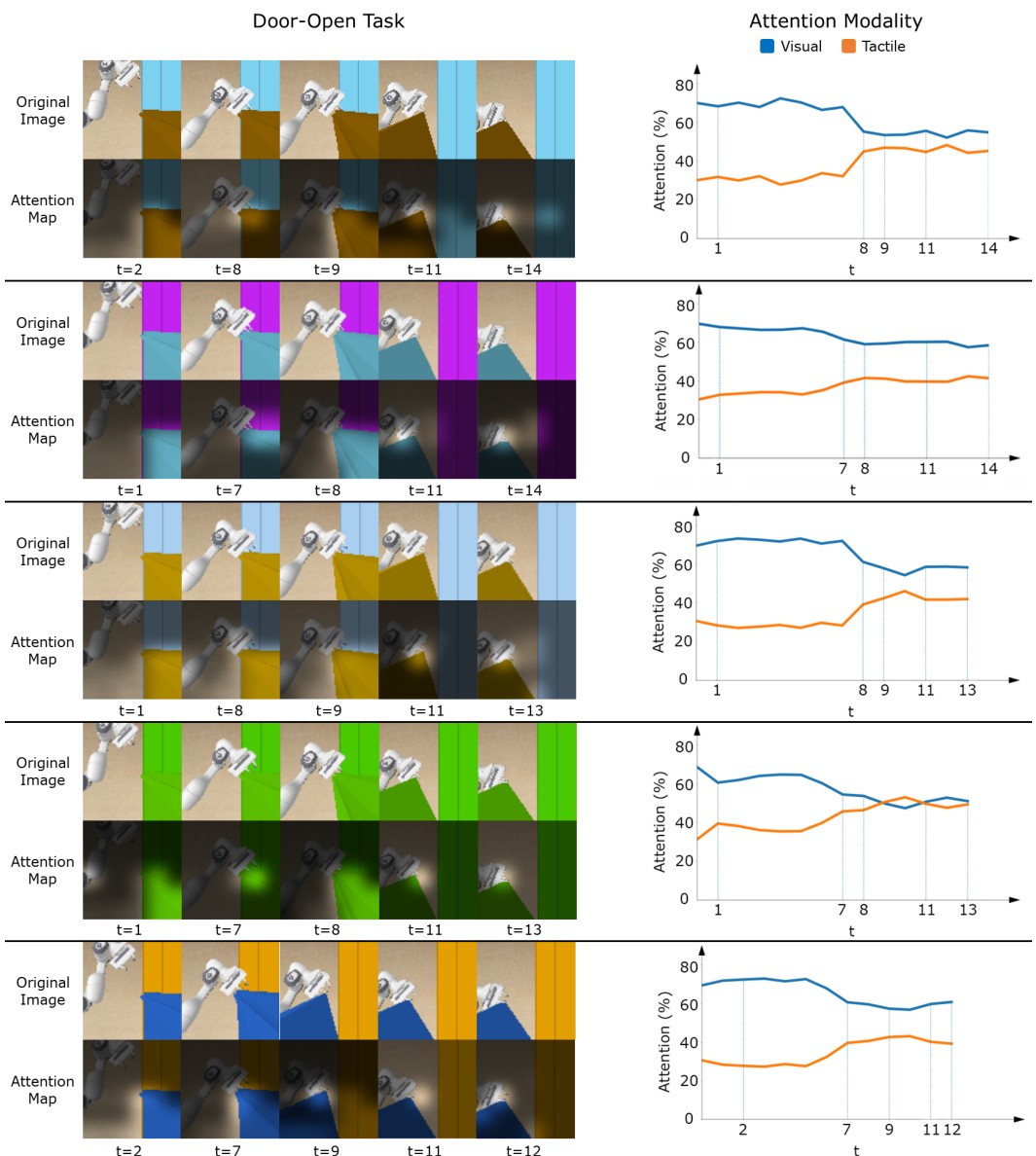

Figure 9: **Attention Visualization for Simulated Tasks:** On the left side, we overlay the attention heatmaps from Eq. 4 onto RGB inputs for 5 instances of the simulated Door-Open task. On the right side, we plot the visual and tactile attention during the task and note the time steps that correspond to the RGB visualizations on the left.

as robotic manipulation. In general, we think of vision as being a global sensing modality and touch as being a local sensing modality. Thus, it stands to reason that until contact is detected, vision outweighs touch.

## 6.4 Parameter Count

We present the exact parameter counts for each model ablated in Sec. 6.4. This ablation is motivated by VTT having roughly 5-6x parameters as our implementation of the concatenation and PoE baselines. After adjusting each of these baselines to have comparable parameters to VTT, we found that their performance did not improve.

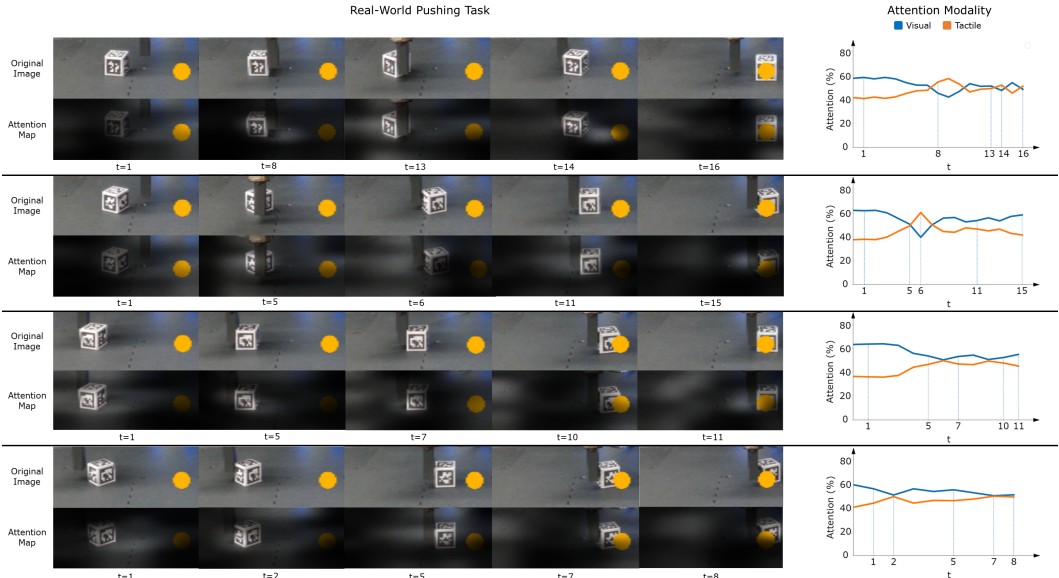

Figure 10: **Attention Visualization for Real-World Tasks:** On the left side, we overlay the attention heatmaps from Eq. 4 onto RGB inputs for 4 instances of the real-world Pushing task. On the right side, we plot the visual and tactile attention during the task and note the time steps that correspond to the RGB visualizations on the left.

Table 1: Parameter counts for all methods.

| Method | Parameters |
|---|---|
| Concatenation | 2.228E5 |
| PoE | 2.889E5 |
| Concatenation w/ Adjustment | 1.110E6 |
| PoE w/ Adjustment | 1.201E6 |
| MulT | 1.118E6 |
| VTT | 1.193E6 |

To ensure that VTT's outperformance of the baselines is not due to the expressiveness of the models, we conduct an ablation study over parameter count. We increase the size of the concatenation and PoE baselines to be comparable to the size of VTT. As we show in Fig. 11, we find that increasing the sizes of concatenation and PoE worsen their performance. Parameter counts are reported in Table 1.

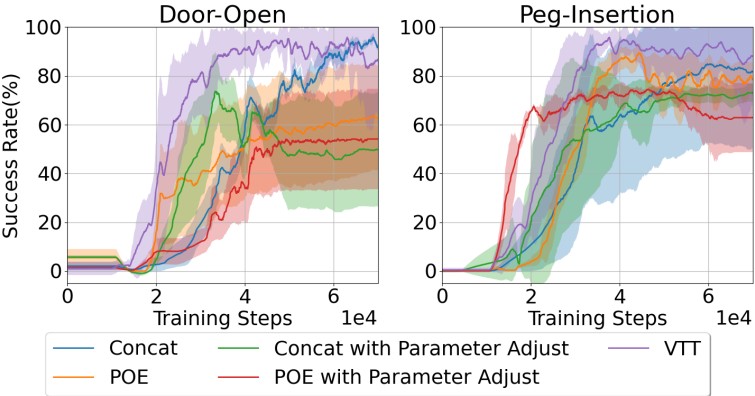

Figure 11: **Parameter Adjustment:** We examine how the size of each model impacts performance and find that larger concatenation and PoE models do not yield better results.

