# OpenReview forum: "Visuo-Tactile Transformers for Manipulation"
_robot-learning.org/CoRL/2022/Conference — CoRL 2022 Poster_

### Official Review · Reviewer_Y7uf · 2022-07-23

**Originality:** Very Good
**Technical Quality:** Very Good
**Clarity Of Presentation:** Excellent
**Impact:** 3

**Recommendation:**

Weak Accept: I recommend accepting the paper, but will not argue for my recommendation if the majority of other reviewers have a different opinion.

**Summary:**

In this paper, the authors propose a multimodal representation learning method named visuo-tactile transformer (VTT) to effectively exploit multimodal feedback from vision and touch. They demonstrate the effectiveness of VTT based on a comparative evaluation against baselines on both simulated and real-world tasks and on an ablation study over the components of VTT.

**Issues:**

Issue 1: The references on page 1 lines 16-17 are not appropriate for reinforcement learning (RL) from raw sensory inputs. The authors need to investigate proper references in the corresponding studies.

Issue 2: The appropriate references should be mentioned for the sentence “most research in robot latent representation learning has focused on images and proprioception” on page 1 line 29.

Issue 3: I suggest the authors conduct an additional ablation study for the compressed latent vector z. For example, the effectiveness of compression and different dimensions of latent vector z.

Issue 4: The authors use SLAC for the RL baseline. Why not Dreamer-v2, currently the state-of-the-art model-based RL method?

Issue 5: The proposed method uses SLAC as an RL baseline. However, the baselines [11, 12] use TRPO as an RL baseline. These two RL algorithms, SLAC and TRPO, are very different; for example, TRPO is a model-free approach whereas SLAC is a model-based approach. It is well-known that SLAC is better sample-efficient than TRPO. The authors need to perform a comparative evaluation against the baselines on the same condition.

**Quality Of The Limitations Section:**

Additional details required

**Reviewer Expertise:**

4: The reviewer is confident but not absolutely certain that the evaluation is correct

**Robotics Focus:**

Sufficient demonstration on hardware

**Strengths And Weaknesses:**

Strengths:
-	The paper is well-written and clearly structured.
-	The contributions are convincing.
-	Multimodal representation learning is getting important in the domain of robotics. The authors propose a multimodal representation learning method suited for reinforcement learning.

Weakness:
-	 The experimental evaluation should be fair. A comparative evaluation against baselines should be conducted in fair conditions.


**Summary Of Recommendation:**

The paper is well-written, easy to read, and clearly structured. Although the main pipeline is the same as the baselines and extended from the prior study, the main contributions of the paper are valid. There are several issues to be addressed though. The issues are described in the issue section.

---

> ### Author Response · Authors · 2022-08-27
> **Response to Reviewer Y7uf**
>
> **Comment:**
>
> We include a .zip file that contains referenced figures and a .pdf of the updated manuscript.
>
> **Comment 1: The references on page 1 lines 16-17 are not appropriate for reinforcement learning (RL) from raw sensory inputs. The authors need to investigate proper references in the corresponding studies.**
>
> We have replaced these references with (Hafter, 2019a; Hafner, 2019b; Lee, 2020; Yarats, 2019).
>
> **Comment 2: The appropriate references should be mentioned for the sentence “most research in robot latent representation learning has focused on images and proprioception” on page 1 line 29.**
>
> We have added several references (Hafter, 2019a; Yarats, 2019; Haarnoja, 2018; Zhang, 2018; Fu, 2021; Torabi, 2019; Cong, 2022) to justify this claim.
>
> **Comment 3: I suggest the authors conduct an additional ablation study for the compressed latent vector z. For example, the effectiveness of compression and different dimensions of latent vector z.**
>
> In response to this comment, we conducted an ablation study for the compressed latent vector z. We found that our method outperforms other compression dimensions, as shown in Supplementary Figure: Z Compression (z_compression.png). We have added these results to the manuscript in Section 4.2.2.
>
> **Comment 4: The authors use SLAC for the RL baseline. Why not Dreamer-v2, currently the state-of-the-art model-based RL method?**
>
> When we attempted to implement Dreamer-v2 based on their published code, we found that using only vision, SLAC outperformed Dreamer as shown in Supplementary Figure: Dreamer vs. SLAC (dreamer_comparison.png). We did test Dreamer-v2 with visual and tactile information, but we found that our implementation did not learn as shown in Supplementary Figure: Initial Dreamer (initial_dreamer.png). Therefore, we implemented our method with SLAC instead of Dreamer-v2.
>
> **Comment 5: The proposed method uses SLAC as an RL baseline. However, the baselines [11, 12] use TRPO as an RL baseline. These two RL algorithms, SLAC and TRPO, are very different; for example, TRPO is a model-free approach whereas SLAC is a model-based approach. It is well-known that SLAC is better sample-efficient than TRPO. The authors need to perform a comparative evaluation against the baselines on the same condition.**
>
> We believe that the reviewer misunderstands our implementation, so we will clarify how we implemented and compared the baselines. The reviewer is correct that the original implementations of baselines [11, 12] use TRPO as an RL baseline. However, in our work, all methods in the comparative evaluation use SLAC as an RL baseline. We replaced TRPO with SLAC in our implementation and evaluation to ensure that all baselines are evaluated under the same conditions.
>
> **References:**
>
> D. Hafner, T. P. Lillicrap, J. Ba, and M. Norouzi. Dream to control: Learning behaviors by latent imagination. CoRR, abs/1912.01603, 2019. URL http://arxiv.org/abs/1912.01603.
>
> D. Hafner, T. Lillicrap, I. Fischer, R. Villegas, D. Ha, H. Lee, and J. Davidson. Learning latent dynamics for planning from pixels. In International Conference on Machine Learning, pages 2555–2565, 2019.
>
> A. X. Lee, A. Nagabandi, P. Abbeel, and S. Levine. Stochastic latent actor-critic: Deep reinforcement learning with a latent variable model. In Advances in Neural Information Processing Systems, volume 33, pages 741–752. Curran Associates, Inc., 2020.
>
> D. Yarats, A. Zhang, I. Kostrikov, B. Amos, J. Pineau, and R. Fergus. Improving sample efficiency in model-free reinforcement learning from images, 10 2019.
>
> T. Haarnoja, A. Zhou, K. Hartikainen, G. Tucker, S. Ha, J. Tan, V. Kumar, H. Zhu, A. Gupta, P. Abbeel, and S. Levine. Soft actor-critic algorithms and applications. CoRR, abs/1812.05905, 2018.
>
> M. Zhang, S. Vikram, L. M. Smith, P. Abbeel, M. J. Johnson, and S. Levine. SO-LAR: deep structured latent representations for model-based reinforcement learning. CoRR, abs/1808.09105, 2018.
>
> Z. Fu, A. Kumar, A. Agarwal, H. Qi, J. Malik, and D. Pathak. Coupling vision and pro-prioception for navigation of legged robots. CoRR, abs/2112.02094, 2021.
>
> F. Torabi, G. Warnell, and P. Stone. Imitation learning from video by leveraging proprioception. CoRR, abs/1905.09335, 2019
>
> L. Cong, H. Liang, P. Ruppel, Y. Shi, M. Gorner, N. Hendrich, and J. Zhang. Reinforcement learning with vision-proprioception model for robot planar pushing. Frontiers in neurorobotics 16:829437, 03 2022.
>
>
> **Zip File:**
>
> /attachment/251f21f1cb7f94d83c4e06d31705a5a58714656f.zip

---

### Official Review · Reviewer_nBFB · 2022-07-29

**Originality:** Good
**Technical Quality:** Good
**Clarity Of Presentation:** Good
**Impact:** 3

**Recommendation:**

Weak Accept: I recommend accepting the paper, but will not argue for my recommendation if the majority of other reviewers have a different opinion.

**Summary:**

This paper proposes the Visuo-Tactile Transformer (VTT) which is a neural network architecture that can be used for model-based reinforcement learning and planning. It takes in visual and tactile information. The tactile information comes from a wrist force-torque sensor. The VTT can take in the RGB image and the force torque sensor information and produce a latent variable z, and that can be the representation used for model-based RL. The paper combines VTT with Stochastic Latent Actor-Critic (SLAC). The paper evaluates on 4 simulated tasks and one physical block pushing task.


**Issues:**

Mostly I have a few clarification questions.

1. In Figure 1, where are the cross-attention layers? Furthermore, can the caption clearly indicate which aspects of this figure are new and which ones are straight from ViT?

2. Regarding VTT and SLAC integration: is VTT trained once and then used as a fixed architecture for getting a latent vector z for SLAC training, or are VTT and SLAC trained together? I think it's the latter, but I hope the paper can clarify (or that the authors can tell me where I missed it in the paper)

3. In Section 4.1, the paper says that the baseline are the methods from Lee et al. However those papers use an optical flow loss. I don't see this in Figure 2. Can the paper clarify this? Otherwise I am concerned that the baseline is not fairly implemented. Also I don't completely buy the claim in lines 218-219 about the peg insertion task evaluated in the baselines, because those prior papers used real world peg insertion, whereas the current paper uses simulation. It feels like this writing should be made more conservative.

4. Lines 212 to 214: the paper says it selects the top 5 of 8 results from each fusion module. I don't understand this comment, is this like taking the top 5 of 8 random seeds?

5. Figures 4 and 5: how many random seeds are part of each curve? Also for Figure 5(b), what happened to the standard deviations? Are these just one trial each (yikes)?

6. I would encourage adding more qualitative examples of the attention heat maps in the supplementary material.

7. Typo: Line 223, "w.r.t. to" should be "w.r.t" or it can be spelled out entirely (there is space in the paper).

8. The BibTeX really needs to be updated, e.g., [35] is a duplicate reference, and [36] through [44] need to have more complete citations. This needs to be cleaned up for the next version.

**Quality Of The Limitations Section:**

Limitations are addressed clearly

**Reviewer Expertise:**

3: The reviewer is fairly confident that the evaluation is correct

**Robotics Focus:**

Sufficient demonstration on hardware

**Strengths And Weaknesses:**

Strengths:

I like the combination of vision and touch. I think learning from multiple sensing modalities is key to unlocking generalist robots. Also, building on Vision Transformer (ViT) and showing good results is ideal as ViT is very popular which might increase the likelihood of other researchers adopting this method.

The experiments seem to be developed under well controlled conditions. In simulation the benchmark is a strong one from prior works, and the paper also has a nice set of ablations. The results from Figures 4 and 5 do seem to show benefits of their method over alternatives or over ablated variants.

The limitations section seems fair and appropriate (though I'd caution, mentioning curiosity is interesting but it feels too speculative to mention this…).

Weaknesses: the robotics tasks seem to be fairy simple though I guess the same could be said for a lot of papers. In the real world the experiments are block pushing, and it seems simplistic.  I don't quite get the feeling that VTT is enabling robots to solve more complex manipulation tasks. Is there a way to address my concern?

There are also a bunch of clarification questions I have, see "issues".


**Summary Of Recommendation:**

I am recommending a weak accept, I think this seems like a reasonably good fit for CoRL and I'm excited to see more vision and tactile work for robotics. I also can't see any glaring drawbacks to the paper at the moment.


**Update after reading author response**

I appreciate the authors taking some of my feedback into account, such as fixing the Figure 2 and adjusting the language and adding some more heat map examples (mostly in supplement). I also like the new ablation on parameter count for the baselines.

Still, I feel a bit hesitant given just 1 real world demonstration tested here (I wish there was more) as they responded to my **Comment 7**. Also from their reply to my **Comment 4** I'm still not sure if it's fair to just use a different loss function to compare against the Lee et al baseline since the choice of loss function could be highly integrated into the choice of the architecture.

Also while I didn't mention this in my initial review, I felt a bit lost reading through Section 3 and the implementation details. I'm not an expert in ViT (the most closely related prior work) so that might have made it harder.

To also clarify: when I was referring to cleaning up the BibTex I was hoping to see things like the conference or journal venue, instead of repeated use of "CoRR".

I originally gave a weak accept rating and I would prefer to keep this rating.

---

> ### Author Response · Authors · 2022-08-27
> **Response to Reviewer nBFB**
>
> **Comment:**
>
> We include a .zip file that contains referenced figures and a .pdf of the updated manuscript.
>
> **Comment 1: The robotics tasks seem to be fairly simple though I guess the same could be said for a lot of papers. In the real world the experiments are block pushing, and it seems simplistic. I don't quite get the feeling that VTT is enabling robots to solve more complex manipulation tasks. Is there a way to address my concern?**
>
> We thank the reviewer for their insight. Block pushing is a fairly common task and is surprisingly complex. For instance, there are a total of 4 dynamics modes at any particular time (out of contact, in contact and sticking, in contact and sliding to left, in contact and sliding to right) – each with its own unique dynamics. Further, the frictional interaction between the block, robot, and surface is complex. For more details please see (Hogan, 2020; Bauza, 2018).
>
> While there certainly are more challenging tasks, we believe this task is on par with many other current works in learning for robotic manipulation and the simulated tasks are an RL suite benchmark.
>
> **Comment 2: In Figure 1, where are the cross-attention layers? Furthermore, can the caption clearly indicate which aspects of this figure are new and which ones are straight from ViT?**
>
> In response to this comment, we have updated Figure 1 (vtt_architecture.png) to highlight our novel contributions and the cross-attention layers.
>
> **Comment 3: Regarding VTT and SLAC integration: is VTT trained once and then used as a fixed architecture for getting a latent vector z for SLAC training, or are VTT and SLAC trained together? I think it's the latter, but I hope the paper can clarify (or that the authors can tell me where I missed it in the paper)**
>
> The reviewer is correct in that VTT and SLAC are trained together. This has been clarified in lines 211-212 of the updated manuscript.
>
> **Comment 4: In Section 4.1, the paper says that the baseline are the methods from Lee et al. However those papers use an optical flow loss. I don't see this in Figure 2. Can the paper clarify this? Otherwise I am concerned that the baseline is not fairly implemented.**
>
> The reviewer is correct that we do not implement the optical flow loss from Lee et al. The purpose of this paper is to compare learning architectures, rather than full loss structures. Therefore, we did not implement this loss.
>
> **Comment 5: Also I don't completely buy the claim in lines 218-219 about the peg insertion task evaluated in the baselines, because those prior papers used real world peg insertion, whereas the current paper uses simulation. It feels like this writing should be made more conservative.**
>
> The observation made in this section of the paper was purely speculative, intended to explore why this baseline outperformed VTT in only the Peg-Insertion task. We are not making any hard claims about the results, and have softened the language in the manuscript.
>
> **Comment 6: Lines 212 to 214: the paper says it selects the top 5 of 8 results from each fusion module. I don't understand this comment, is this like taking the top 5 of 8 random seeds?**
>
> We have updated the manuscript to include all 8 results in the final version. We took this approach in the initial version because we noted that many works include exactly 5 results (Igl, 2018; Colas, 2018; Hafner, 2018; Haarnoja, 2018; Hafner, 2020).
>
> **Comment 7: Figures 4 and 5: how many random seeds are part of each curve? Also for Figure 5(b), what happened to the standard deviations? Are these just one trial each (yikes)?**
>
> In Figure 4 and Figure 5(a), the curves show the top 5 seeds. We have updated Figure 4 (sim_results.png) and Figure 5 (ablation_results.png) to include all 8 seeds. In Figure 5(b), this is a single trial. In the updated manuscript, we refer to this as a demonstration rather than an experiment due to the limited trials available.
>
> **Comment 8: I would encourage adding more qualitative examples of the attention heat maps in the supplementary material.**
>
> In response to this comment, we have added to the Supplementary Material 5 additional qualitative attention heatmap examples for the simulated Door-Open task and 4 additional qualitative attention heatmap examples for the real-world Block Pushing task in Supplementary Figures: Simulated Attention Visualization (sim_attn_viz.png) and Real-World Attention Visualization (rw_attn_viz.png). Additionally, we have included plots for each that show the balance of visual and tactile attention at each of the visualized time steps.
>
> **Comment 9: Line 223, "w.r.t. to" should be "w.r.t" or it can be spelled out entirely (there is space in the paper).**
>
> This correction has been made.
>
> **Comment 10: The BibTeX really needs to be updated, e.g., [35] is a duplicate reference, and [36] through [44] need to have more complete citations. This needs to be cleaned up for the next version.**
>
> The references section has been updated accordingly.
>
>
> **Zip File:**
>
> /attachment/f8e60b0b15a174f032aab173f2a42faa73bd3821.zip

---

> ### Author Response · Authors · 2022-08-27
> **Response to Reviewer nBFB (continued)**
>
> References:
>
> Hogan, F. R., & Rodriguez, A. Feedback control of the pusher-slider system: A story of hybrid and underactuated contact dynamics. In Algorithmic Foundations of Robotics XII (pp. 800-815). Springer, Cham. 2020.
>
> Bauza, M., Hogan, F. R., & Rodriguez, A. A data-efficient approach to precise and controlled pushing. In Conference on Robot Learning (pp. 336-345). PMLR. 2018.
>
> M. Igl, L. Zintgraf, T. A. Le, F. Wood, and S. Whiteson, “Deep variational reinforcement learning for POMDPs,” in Proceedings of the 28th International Conference on machine learning (ICML), 2018.
>
> Cédric Colas, Olivier Sigaud, and Pierre-Yves Oudeyer. How many random seeds? statistical power analysis in deep reinforcement learning experiments. arXiv preprint arXiv:1806.08295, 2018.
>
> Hafner, D., Lillicrap, T., Fischer, I., Villegas, R., Ha, D., Lee, H., and Davidson, J. (2018). Learning latent dynamics for planning from pixels. arXiv preprint arXiv:1811.04551.
>
> Haarnoja, T., Zhou, A., Abbeel, P., and Levine, S. Soft actor-critic: Off-policy maximum entropy deep reinforcement learning with a stochastic actor. In International Conference on Machine Learning (ICML), 2018c.
>
> Danijar Hafner, Timothy Lillicrap, Mohammad Norouzi, and Jimmy Ba. Mastering atari with discrete world models. arXiv preprint arXiv:2010.02193, 2020.

---

### Official Review · Reviewer_2Auu · 2022-07-29

**Originality:** Good
**Technical Quality:** Fair
**Clarity Of Presentation:** Good
**Impact:** 4

**Recommendation:**

Weak Accept: I recommend accepting the paper, but will not argue for my recommendation if the majority of other reviewers have a different opinion.

**Summary:**

The paper introduces a multimodal representation learning method which combines visual and tactile information using a transformer architecture, to act as a latent representation learning model in methods such as SLAC (stochastic latent actor critic). The method is evaluated on several simulated robotic environments in which the observations are RGB images as well as signals from a force-torque sensor, and also on a real world block pushing task. Compared to other fusion baselines like naive concatenation and a product of experts, the proposed method is able to achieve better performance and sample efficiency on the studied tasks.


**Issues:**

- What considerations are made to make the baseline comparisons as fair as possible in terms of parameter count? It could be that the performance improvements of the transformer method comes from simply having a more expressive model.
- Experimental results are reported on the top 5 of 8 results for each method. I don’t believe this is a standard evaluation protocol so it would be good to see the full results or understand why this choice was made.
- I assume that, as in Lee et. al, the alignment embedding is used alongside a training scheme which samples both “aligned” data as well as temporally “nonaligned” data, but I don’t think this is mentioned in the manuscript.
- The introduction argues that “recent research in representation learning aims to build compact “latent” representations of the underlying task”, but I would argue that the representations are of the states rather than the task itself.


**Quality Of The Limitations Section:**

Limitations are addressed clearly

**Reviewer Expertise:**

3: The reviewer is fairly confident that the evaluation is correct

**Robotics Focus:**

Sufficient demonstration on hardware

**Strengths And Weaknesses:**

Strengths:
- The paper is well motivated and timely – the problem of how to effectively combine visual and tactile information for robot learning is one which is beginning to be explored but still quite open.
- The idea of using attention blocks and a transformer architecture for this fusion is quite promising due to the success of methods like vision transformers.
- The experimental results are quite strong. Although I have a few concerns about the evaluation protocol as mentioned later in the review, the results presented in Figure 4 do show substantial improvement over concatenation and a strong previous method in PoE.
- The qualitative examples of visualized attention maps are quite convincing that the method does indeed learn to focus on the contact interactions when present.

Weaknesses:
- The presentation of the main method in section 3 makes it slightly confusing for me to understand. From my understanding, it seems like the cross-modal attention is achieved by performing self attention using a set of inputs which contains both the embedded modality patches of images as well as tactile sensing information. The demonstration of the image and tactile self and cross-modal attention in Equation 4 is nice, but many symbols like the $A_{n=i}^{self}$ are introduced and only used a couple of times which makes it hard to keep track of the computation. Additionally, I’m not sure there is a large benefit from describing the attention fusion layers as iterations rather than simply layers? The form of Equation 7 doesn’t add anything for me in addition to the previous description.
- Particularly, I think the learned embeddings introduced in section 3.3 are a bit confusing – my understanding that a contact embedding $X_C$ is learned, not dependent on any of the image or tactile inputs, and a ground truth contact state is used to supervise the output at that position after attention layers are applied. A nitpick is that the phrasing “After N iterations, the contact embedding becomes the contact head” is rather confusing because it suggests that the contact head is somehow set to the contact embedding vector, whereas the contact head (I believe) is rather an entirely different vector, the output of a stack of attention layers.


**Summary Of Recommendation:**

Generally, I feel that the method is well motivated and achieves strong results. While the overall novelty of the method is somewhat limited, I think it is a nice step in the direction of multimodal fusion for robot learning. I will definitely consider raising my score if the authors address the issues particularly with evaluation in the “issues” section below.

---

> ### Author Response · Authors · 2022-08-27
> **Response to Reviewer 2Auu**
>
> **Comment:**
>
> We include a .zip file that contains referenced figures and a .pdf of the updated manuscript.
>
> **Comment 1: The presentation of the main method in section 3 makes it slightly confusing for me to understand. From my understanding, it seems like the cross-modal attention is achieved by performing self attention using a set of inputs which contains both the embedded modality patches of images as well as tactile sensing information. The demonstration of the image and tactile self and cross-modal attention in Equation 4 is nice, but many symbols like the An=iself are introduced and only used a couple of times which makes it hard to keep track of the computation. Additionally, I’m not sure there is a large benefit from describing the attention fusion layers as iterations rather than simply layers? The form of Equation 7 doesn’t add anything for me in addition to the previous description.**
>
> The reviewer’s understanding of the main method is correct. We have updated our manuscript to clean up Equations 4 and 6 and to remove Equation 7.
>
> **Comment 2: Particularly, I think the learned embeddings introduced in section 3.3 are a bit confusing – my understanding that a contact embedding XC is learned, not dependent on any of the image or tactile inputs, and a ground truth contact state is used to supervise the output at that position after attention layers are applied. A nitpick is that the phrasing “After N iterations, the contact embedding becomes the contact head” is rather confusing because it suggests that the contact head is somehow set to the contact embedding vector, whereas the contact head (I believe) is rather an entirely different vector, the output of a stack of attention layers.**
>
> The first output is indeed a stack of attention layers, but we compress them with sigmoid functions to classify the contact and alignment heads. We have updated Figure 1 (vtt_architecture.png) to provide more clarity. We have updated our language to be consistent with using “layers” instead of “iterations” throughout Sec. 3.
>
> **Comment 3: What considerations are made to make the baseline comparisons as fair as possible in terms of parameter count? It could be that the performance improvements of the transformer method comes from simply having a more expressive model.**
>
> After analyzing our method and the baselines, we found that VTT has more parameters than either baseline. In order to ensure that our comparison results were fair, we increased the number of parameters in each baseline to be comparable with the number of parameters in VTT. We show these results in Supplementary Figure: Parameter Ablation (param_ablation.png) and note that they do not improve the results of the baseline. Therefore, we believe that our results from the initial version stand and are fair. We have added these results to the manuscript in Section 4.2.3.
>
> **Comment 4: Experimental results are reported on the top 5 of 8 results for each method. I don’t believe this is a standard evaluation protocol so it would be good to see the full results or understand why this choice was made.**
>
> We have updated the manuscript to include all 8 results in the final version. We took this approach in the initial version because we noted that many works include exactly 5 results (Igl, 2018; Colas, 2018; Hafner, 2018; Haarnoja, 2018; Hafner, 2020).
>
> **Comment 5: I assume that, as in Lee et. al, the alignment embedding is used alongside a training scheme which samples both “aligned” data as well as temporally “nonaligned” data, but I don’t think this is mentioned in the manuscript.**
>
> This assumption is correct. We have updated the manuscript to note it in lines 443-445.
>
> **Comment 6: The introduction argues that “recent research in representation learning aims to build compact “latent” representations of the underlying task”, but I would argue that the representations are of the states rather than the task itself.**
>
> This has been corrected in the manuscript (line 22).
>
> **References:**
>
> M. Igl, L. Zintgraf, T. A. Le, F. Wood, and S. Whiteson, “Deep variational reinforcement learning for POMDPs,” in Proceedings of the 28th International Conference on machine learning (ICML), 2018.
>
> Cédric Colas, Olivier Sigaud, and Pierre-Yves Oudeyer. How many random seeds? statistical power analysis in deep reinforcement learning experiments. arXiv preprint arXiv:1806.08295, 2018.
>
> Hafner, D., Lillicrap, T., Fischer, I., Villegas, R., Ha, D., Lee, H., and Davidson, J. (2018). Learning latent dynamics for planning from pixels. arXiv preprint arXiv:1811.04551.
>
> Haarnoja, T., Zhou, A., Abbeel, P., and Levine, S. Soft actor-critic: Off-policy maximum entropy deep reinforcement learning with a stochastic actor. In International Conference on Machine Learning (ICML), 2018c.
>
> Danijar Hafner, Timothy Lillicrap, Mohammad Norouzi, and Jimmy Ba. Mastering atari with discrete world models. arXiv preprint arXiv:2010.02193, 2020.
>
>
> **Zip File:**
>
> /attachment/b63864f6903275bef47aed411e9d8bd497bfd85c.zip

---

### Official Review · Reviewer_KWDi · 2022-07-31

**Originality:** Fair
**Technical Quality:** Good
**Clarity Of Presentation:** Good
**Impact:** 3

**Recommendation:**

Weak Accept: I recommend accepting the paper, but will not argue for my recommendation if the majority of other reviewers have a different opinion.

**Summary:**

The paper extends vision transformers and proposes visuo-tactile transformer to perform modality fusion, and demonstrate the effectiveness of the proposed architecture on four simulated robot tasks and one real-world block pushing task, outperforming two alternative fusion methods.

**Issues:**

Please see the discussion in Strengths And Weaknesses

**Quality Of The Limitations Section:**

Limitations are addressed clearly

**Reviewer Expertise:**

4: The reviewer is confident but not absolutely certain that the evaluation is correct

**Robotics Focus:**

Sufficient demonstration on hardware

**Strengths And Weaknesses:**

Strengths:

- The proposed approach extends the popular Visual Transformer (ViT) [1] to integrate multimodal feedback from vision and touch – the Visuo-Tactile Transformer (VTT), which is well-motivated.

- On both simulated data and real-world experiments, the proposed Visual-Tactile Transformer architecture outperforms two alternative fusion methods: simple concatenation and Product-of-Experts.

- Qualitative analysis demonstrates the attention modules in the transformer architecture successfully attends to the important regions to achieve the tasks.

Weaknesses:

- The paper claims to perform visuo-tactile representation learning, which is somewhat confusing. Throughout the paper, the main focus is more on using VTT to perform modality fusion instead of doing representation learning.

- The proposed VTT architecture is very similar to ViT[1] with some minimal changes. The main contribution/idea is to also divide for-ce-torque data into temporal patches and use the same vision transformer architecture to process and fuse the two modalities. It is important and necessary to highlight the key differences of the proposed architecture compared to ViT (be explicit on which part is from ViT and which part is new), and show why the proposed new components are necessary instead of just applying it to tactile data.

- For qualitative results, the attention map is only shown and overlaid on visual images. However, the key idea is attention on tactile data, therefore qualitative analysis of tactile data would be more useful and important. Otherwise, it's very much expected because the architecture for the vision part is similar to or almost the same as ViT.





**Summary Of Recommendation:**

The paper extends vision transformer (ViT) and introduces VTT that also uses tactile data. Most of the network design is the same as ViT. Though the proposed approach outperforms two alternative fusion methods, the novelty of the method is somewhat limited. And the main contribution is adding tactile data (force and torque) to ViT and showing it's better than two other fusion methods. Therefore, I am leading towards rejection of the paper but happy to discuss more in the rebuttal discussion.

---

> ### Author Response · Authors · 2022-08-27
> **Response to Reviewer KWDi**
>
> **Comment:**
>
> We include a .zip file that contains referenced figures  and a .pdf of the updated manuscript.
>
> **Comment 1: The paper claims to perform visuo-tactile representation learning, which is somewhat confusing. Throughout the paper, the main focus is more on using VTT to perform modality fusion instead of doing representation learning.**
>
> VTT is an approach to multimodal representation learning that fuses vision and touch to produce a lower-dimensional representation. In essence, we are doing a type of representation learning that focuses on multimodal fusion.
>
> **Comment 2: The proposed VTT architecture is very similar to ViT[1] with some minimal changes. The main contribution/idea is to also divide force-torque data into temporal patches and use the same vision transformer architecture to process and fuse the two modalities. It is important and necessary to highlight the key differences of the proposed architecture compared to ViT (be explicit on which part is from ViT and which part is new), and show why the proposed new components are necessary instead of just applying it to tactile data.**
>
> In response to this comment, we have updated Figure 1 (vtt_architecture.png) to more clearly show our novel contributions to the architecture. In addition to introducing tactile data, we introduce the alignment and contact embeddings and heads. These are beneficial for cross-attention formulation for robotics applications that utilize both vision and touch. We show the necessity of these embeddings and losses in the ablation study conducted in Section 4.2.1. From a theoretical perspective, these embeddings compact the latent space in a meaningful way: as shown in Supplementary Figure: Embeddings (embeddings.png), the contact embedding brings contact states closer together in the latent space while pushing non-contact states apart. For the alignment embedding, the same thing is being done. Temporally misaligned states are being pushed apart in the latent space while temporally aligned latent states are being drawn together. Intuitively, applying ViT directly to tactile data would not be effective because the visual component is important in several ways: take the block pushing task as an example. Without vision, identifying the start state of the block, identifying the state of the robot itself, and qualifying when the block has reached the goal state would be inefficient at best and impossible at worst with purely tactile information.
>
> **Comment 3: For qualitative results, the attention map is only shown and overlaid on visual images. However, the key idea is attention on tactile data, therefore qualitative analysis of tactile data would be more useful and important. Otherwise, it's very much expected because the architecture for the vision part is similar to or almost the same as ViT.**
>
> To address this comment, we additionally visualize the proportion of visual and tactile attention over time during the simulated Door-Open task and the real-world Block Pushing task in the attached Supplementary Figures: Simulated Attention Visualization (sim_attn_viz.png) and Real-World Attention Visualization (rw_attn_viz.png). We find that during contact, the tactile attention is higher than when no contact is present. We find this result to be unsurprising because it is in line with our intuition about the nature of multimodality in contact-rich settings such as robotic manipulation. In general, we think of vision as being a global sensing modality and touch as being a local sensing modality. Thus, it stands to reason that until contact is detected, vision outweighs touch. We also notice that attention to vision and touch are more balanced in the real-world tasks than in the simulated tasks. We have added Section 6.3 to the appendix to discuss these findings in greater detail.
>
> **Comment 4: The paper extends vision transformer (ViT) and introduces VTT that also uses tactile data. Most of the network design is the same as ViT. Though the proposed approach outperforms two alternative fusion methods, the novelty of the method is somewhat limited. And the main contribution is adding tactile data (force and torque) to ViT and showing it's better than two other fusion methods. Therefore, I am leading towards rejection of the paper but happy to discuss more in the rebuttal discussion.**
>
> In response to this comment, we would like to point out that the current arsenal of multimodal fusion techniques in robotics is quite limited and the methods are generally very simple. Therefore, we believe that our contribution is important because we’re taking a state-of-the-arm method from the Natural Language Processing/Computer Vision community and adapting it to robotics applications. We believe that these kinds of extensions are necessary to solve important open problems in robotics. For more information, please refer to “Response to All Reviewers.”
>
> **Zip File:**
>
> /attachment/8ca651a5a82b9cc1f8a898c5fcc327ca1b18d00a.zip

---

### Author Response · Authors · 2022-08-27
**Response to All Reviewers**

The authors would like to thank the reviewers for taking the time to provide insightful and thorough comments on our work. The reviewers have stated that our paper is “enjoyable to read”, the experimental results are “quite strong”, and the problem is “well-motivated and timely.” Here, we summarize our major responses:

The most common concern in the reviews was whether or not the baselines were being compared fairly with our method. Based on this feedback, we took two actions. Firstly, we conducted an ablation of parameter count. We increased the parameter count of both the Concatenation and Product of Experts baselines to be comparable with VTT. We found that these adjustments did not significantly change our results, so we believe that our results hold and are fair. Secondly, we present all 8 of our trial results rather than the top 5. We found that this did not significantly impact our results, but we originally showed these selected results because we had seen several papers that presented exactly 5 results (Igl, 2018; Colas, 2018; Hafner, 2018; Haarnoja, 2018; Hafner, 2020).

Reviewers KWDi and 2Auu had concerns that the novelty of our method is somewhat limited. In response to these comments, we would like to point out that the current arsenal of multimodal fusion techniques in robotics is quite limited and the methods are generally very simple. Therefore, we believe that our contribution is important because we’re taking a state-of-the-art method from the Natural Language Processing/Computer Vision community and adapting it to robotics applications. We believe that these kinds of extensions are necessary to solve important open problems in robotics.

Reviewers suggested additional references and minor corrections for existing references, which we have taken.

Reviewers also suggested minor spelling and grammatical corrections, which we have taken.

We respond to other specific comments in the replies to reviewers below. We have included an updated manuscript with all edits in blue text. Again, we thank the reviewers for their detailed comments.

References:

M. Igl, L. Zintgraf, T. A. Le, F. Wood, and S. Whiteson, “Deep variational reinforcement learning for POMDPs,” in Proceedings of the 28th International Conference on machine learning (ICML), 2018.

Cédric Colas, Olivier Sigaud, and Pierre-Yves Oudeyer. How many random seeds? statistical power analysis in deep reinforcement learning experiments. arXiv preprint arXiv:1806.08295, 2018.

Hafner, D., Lillicrap, T., Fischer, I., Villegas, R., Ha, D., Lee, H., and Davidson, J. (2018). Learning latent dynamics for planning from pixels. arXiv preprint arXiv:1811.04551.

Haarnoja, T., Zhou, A., Abbeel, P., and Levine, S. Soft actor-critic: Off-policy maximum entropy deep reinforcement learning with a stochastic actor. In International Conference on Machine Learning (ICML), 2018c.

Danijar Hafner, Timothy Lillicrap, Mohammad Norouzi, and Jimmy Ba. Mastering atari with discrete world models. arXiv preprint arXiv:2010.02193, 2020.

---

### Meta-Review · Area_Chair_x23C · 2022-08-09

**Recommendation:** Accept (Poster)
**Confidence:** 4

**Metareview:**

This work investigated the multimodal fusion of visuo-tactile sensory data with transformer-based model architectures. Their approach used self and cross-attention mechanisms in transformers to build latent heat map representations and applied the learned representations for manipulation tasks in simulation and on real hardware. The idea of combining vision and touch has been well explored in the robotics literature. This work's main contribution is to explore the new attention-based model architectures and show their effectiveness in learning better multimodal representations. This paper received mixed initial reviews with two Weak Rejects and two Weak Accepts. The reviewers appreciated the importance of the problem and the results and analyses showing the advantage of the proposed approach. Meanwhile, they also thought that the technical novelty was somewhat limited (the model is based on ViT with minor changes) and that the evaluation tasks, especially the real-world tasks, were simplistic. Furthermore, three reviewers (Y7uf, nBFB, 2Auu) raised concerns about the fair comparisons between the proposed approach and baselines. The AC would like to see the authors address the comments of all reviewers and look forward to seeing the authors' responses in the discussion phase.

**Post-rebuttal updates:** This paper received mixed ratings of two Weak Accepts and two Weak Rejects at initial reviews. The authors did a great job in their rebuttal. The new ablation studies among other changes in the paper have successfully convinced two reviewers to update their scores from Weak Reject to Weak Accept. Considering the interesting results demonstrated in the paper as well as the overall support of the reviewers, the AC would recommend accepting this paper for CoRL.

**Best Paper Nomination:**

No

---

> ### Author Response · Authors · 2022-08-27
> **Response to Area Chair x23C**
>
> The AC requested that we address the comments of all reviewers, specifically noting the comments pertaining to technical novelty, evaluation tasks, and fair comparisons between our approach and baselines. We have extracted and addressed all weaknesses, issues, and comments to the best of our ability, with significant attention given to ensuring fair comparisons.